# Long-Term Effect of Honeycomb β-Tricalcium Phosphate on Zygomatic Bone Regeneration in Rats

**DOI:** 10.3390/ma13235374

**Published:** 2020-11-26

**Authors:** Ryoko Nakagiri, Satoko Watanabe, Kiyofumi Takabatake, Hidetsugu Tsujigiwa, Toshiyuki Watanabe, Hiroshi Matsumoto, Yoshihiro Kimata

**Affiliations:** 1Department of Plastic and Reconstructive Surgery, Graduate School of Medicine, Dentistry and Pharmaceutical Sciences, Okayama University, Okayama 700-8558, Japan; nakagiriryoko@gmail.com (R.N.); watanabetoshiii@gmail.com (T.W.); ojyarumatsu@ninus.ocn.ne.jp (H.M.); ykimata@cc.okayama-u.ac.jp (Y.K.); 2Department of Oral Pathology and Medicine, Graduate School of Medicine, Dentistry and Pharmaceutical Sciences, Okayama University, Okayama 700-8558, Japan; gmd422094@s.okayama-u.ac.jp; 3Department of Life Science, Faculty of Science, University Science, Okayama 700-0005, Japan; tsuji@dls.ous.ac.jp

**Keywords:** bioceramic, bone tissue regeneration, honeycomb β-TCP, bone remodeling, reproduction of morphology

## Abstract

In recent years, artificial bones with high biocompatibility have been developed for hard tissue reconstruction. However, current bone replacement methods are inadequate for large defects, causing infection, exposure, and damage. We have developed a new honeycomb β-tricalcium phosphate (TCP) material, which achieved good bone regeneration after implantation in a rat complete zygomatic bone defect. In this study, we further investigated the ability of honeycomb β- TCP for remodeling after bone regeneration as a long-term result. Bone morphogenic protein (BMP)-2-free honeycomb β-TCP (TCP group) and honeycomb β-TCP with BMP-2 (BMP group) were implanted in the zygomatic bone of rats. Micro-computed tomography was performed to track the zygomatic bone morphology, and specimens were histologically examined for osteogenesis and remodeling. In the TCP group, no bone formation was observed at 1 month, but it was observed at 6 months. Bone formation was observed in the BMP group at 1 month, and β-TCP absorption reproducing the zygomatic bone morphology was observed at 6 months. This honeycomb β-TCP with BMP-2 may provide appropriate remodeling that reproduces good bone formation in the early stage and good morphology in the long term, offering an alternative bone reconstruction material to vascularized bone grafts.

## 1. Introduction

Large bone defects caused by malignant tumor resection, trauma, and infection are currently reconstructed using autologous bone or biomaterials. Good bone union can be achieved using a vascularized autologous bone graft [1,2,3,4,5,6,7]. However, autologous bone grafts involve major invasion of the donor site. Furthermore, implantation of a vascularized bone graft requires a long operation time and may cause serious complications, including total necrosis of the transplanted tissue due to thrombus of the feeding blood vessel [7,8]. Alternatively, the use of synthetic biomaterials as bone grafts does not require a donor site and enables performing shorter and less invasive surgeries. Various biomaterials have been developed that can overcome the shortcomings of autologous bone transplantation [9,10]. Nevertheless, because biomaterials are foreign bodies, the main limitations of their use include increased risks of long-term infection, exposure, and damage. At present, metals such as titanium or bioceramics including hydroxyapatite (HAp) and tricalcium phosphate (TCP), are widely used as biomaterials for bone grafts. Titanium has relatively good biocompatibility along with excellent workability and strength but is not bioabsorbable and is considered to be a foreign substance by the body. On the other hand, HAp and TCP have inferior strength to metals, but have osteoconductivity and excellent biocompatibility. These materials allow for the migration, proliferation, differentiation, and extracellular matrix (ECM) deposition of bone progenitor cells. Furthermore, HAp is hardly absorbed in vivo, and absorption is very slow if it does occur. In contrast, TCP is easily decomposed and absorbed in vivo. However, if bone formation in the central part of the biomaterial is insufficient, complete replacement with new bone tissue is hindered. The material then continues to be considered a foreign substance, and sufficient strength cannot be achieved. Consequently, to allow for adaptation to areas with low blood supply (such as large bone defects), an ideal scaffold that facilitates bone cell and blood vessel invasion at the interior of the material is desired. 

Several recent studies have suggested that the geometry of biomaterials is important for inducing cell differentiation and tissue formation. To induce hard tissue formation, the optimum shape and size of the pores and optimum geometric structure have been determined [11,12,13,14]. Kuboki et al. [11] showed that straight tunnel-shaped HAp with a pore size of 300–400 μm achieved good bone formation. They further determined that the structure of the open tube induces better angiogenesis and bone formation than that of the dead-end tube. Therefore, we postulated that the development of TCP with a honeycomb structure would provide good bone regeneration to the center of the defect, and that TCP decomposition and resorption would enable complete bone replacement. 

We have succeeded in developing a new honeycomb β-TCP scaffolding material with through-and-through holes, and it was found that sintering at 1200 °C yields high biocompatibility and bone conductivity [15]. We also showed that bone formation can be controlled by the size of the through-and-through holes and amount of bone morphogenic protein (BMP)-2. The material with 300 μm through-and-through holes provided an environment suitable for cell proliferation and differentiation and was optimal for inducing bone tissue regeneration [16]. Furthermore, we obtained good bone tissue regeneration in vivo after implanting honeycomb β-TCP with 300 μm through-and-through holes in a rat zygomatic bone defect model. Although bone tissue regeneration was confirmed, long-term observation was not performed, and thus details regarding the maintenance of regenerated bone tissue, bone remodeling, and β-TCP absorption are unknown. In these previous studies, we used Matrigel for BMP-2 addition, which is often used for culturing induced pluripotent stem cells. However, as Matrigel is comprised of ECM extracted from mouse cancer tissue, its clinical application in humans is difficult. Therefore, in this study, we used the same honeycomb β-TCP with 300 μm through-and-through holes as a bone reconstruction material in a zygomatic bone defect model to observe the long-term result. In addition, we evaluated whether honeycomb β-TCP is suitable as a bone reconstruction material for clinical application, using both Matrigel and collagen, the latter of which is already used clinically as ECM.

## 2. Materials and Methods 

### 2.1. Preparation of Honeycomb β-TCP Scaffolds Containing BMP-2

Honeycomb β-TCP sintered at 1200 °C with a diameter of 3 mm, length of 5 mm, and containing 37 through-and-through holes of 300 μm diameter was purchased from Pilot Corporation (Tokyo, Japan) [14,15]. The honeycomb β-TCP was autoclaved immediately before implantation for the experiment (Figure 1a). For the BMP-2 concentration in this study, we determined the concentration that showed the most bone formation in our previous study [16]. BMP-2 (Peprotech, NJ, USA) was prepared and diluted with BD Matrigel™ (BD Biosciences, Bedford, MA, USA; hereafter referred to as Matrigel) or AteloCell^®^ (KOKEN, Tokyo, Japan; hereafter referred to as collagen gel) to a concentration of 80 μg/mL. The BMP-2 solution (12.5 μL; 1 μg BMP-2) was filled into the β-TCP scaffold by centrifuging at 1500 rpm for 1 min and then incubated at 37 °C for 30 min. In the group implanted with β-TCP alone, the honeycomb β-TCP was filled with 12.5 μL of Matrigel or collagen gel without BMP-2.

### 2.2. Animals and Implantation Procedure

All experiments in this study were approved by the Animal Care and Use Committee, Okayama University (OKU-2019493). All surgical procedures were performed under general anesthesia. Five-week-old healthy male Wistar rats (Charles River Laboratories Japan, Kanagawa, Japan) were used in the experiment. The rats were anesthetized by intraperitoneal injection of ketamine hydrochloride (75 mg/kg) and medetomidine hydrochloride (0.5 mg/kg). After operation, atipamezole hydrochloride (1 mg/kg) was intraperitoneally injected to awaken.

For implantation, a skin incision of approximately 8 mm was made at the anterior site of the zygomatic arch, and the zygomatic periosteum was exposed. Next, the zygomatic periosteum was incised with a surgical knife and completely peeled from the zygomatic bone. The zygomatic arch was cut with scissors to produce a complete bone defect of 5 mm. Subsequently, β-TCP alone (TCP group) or β-TCP with BMP-2 added (BMP group) was implanted in the bone defect. In all cases, β-TCP was implanted so that the anterior part was in contact with the zygomatic body, the posterior part was in contact with the zygomatic arch, and the through-and-through holes of β-TCP and the long axis of the bone defect were parallel to each other (Figure 1b,c). 

### 2.3. Implantation Using Matrigel and Collagen Gel

Matrigel was used as the carrier for BMP-2 in this experiment. In addition to the TCP and BMP groups, a zygomatic bone defect group in which β-TCP was not implanted was also established as a control group. Only the left side was operated, and 6 months after the operation, the rats were euthanized with carbon dioxide gas. Specimens were then collected and fixed with 4% paraformaldehyde (PFA) (control group: n = 3, TCP group: n = 3, BMP group: n = 4).

Collagen gel was also used as the carrier for BMP-2 in this experiment. Based on the results of implantation using Matrigel, implantation using collagen gel as an alternative carrier were performed in the TCP and BMP groups to observe the long-term fate of β-TCP. For micro-CT imaging, only the left zygomatic bone was operated on, and β-TCP was implanted in the same manner as Matrigel experiment (TCP group: n = 4, BMP group: n = 4). In addition, β-TCP was implanted in the zygomatic bone of another group of rats for histological evaluation 1 month after implantation (TCP group: n = 2, BMP group: n = 2).

### 2.4. Micro-Computed Tomography (CT)

After fixation, the heads of the rats were photographed with a micro-CT LaTheta LCT200 system (Hitachi Aloka Medical, Tokyo, Japan), and the DICOM data obtained were reconstructed using the AZE Virtual Place Lexus 64 workstation and software (AZE, Tokyo, Japan). The zygomatic bone morphology was then qualitatively evaluated from the image in Matrigel experiment at 6 months after the operation.

In collagen gel experiments, at 1 month, 3 months, and 6 months after the operation, we performed micro-CT of the heads of the rats under general anesthesia (isoflurane inhalation) to track the course of recovery and bone regeneration following the operation. Bone morphology was qualitatively evaluated using three-dimensional (3D) reconstructed images as in the Matrigel experiment. Furthermore, the volume of β-TCP and the cross-sectional area at the rear end were measured and evaluated quantitatively.

The volume of β-TCP was measured using LaTheta software v.3.50 (Hitachi Aloka Medical, Tokyo, Japan) by designating the β-TCP portion as the region of interest for each slice (Figure 2a,b). The β-TCP residual rate (%) was calculated from the volumes at 1 month, 3 months, and 6 months, assuming that the volume of β-TCP was 100% immediately after implantation.

To evaluate the change of the cross-sectional area of β-TCP, AZE Virtual Place Lexus 64 was used to reconstruct the image of the rear edge of β-TCP in a plane perpendicular to the through-and-through holes (Figure 2c,d). Six months after implantation, the area of β-TCP was measured using ImageJ software v.1.52 (National Institutes of Health, Bethesda, MD, USA) and was calculated as a percentage relative to the area in a cross-section before implantation.

### 2.5. Histology

Rats were euthanized 1 or 6 months after implantation, and tissues containing β-TCP were removed. The excised tissue was fixed with 4% PFA and then decalcified in 10% ethylenediaminetetraacetic acid for 3 weeks. The tissue was embedded in paraffin and sliced to a thickness of 5 μm. The sections were chemically stained with hematoxylin and eosin (HE) and observed histologically.

### 2.6. Immunohistochemical Staining of Osteopontin (OPN)

In the specimens collected at 1 month after the operation, the presence of OPN, a non-collagen protein produced by osteoblasts in the ECM of the bone, was confirmed to evaluate bone formation. 

The sections were deparaffinized, hydrophilized, and then incubated in proteinase K for 15 min at room temperature. Endogenous peroxidase was blocked with a 0.3% hydrogen peroxide solution in methanol for 20 min. Non-specific binding sites were blocked with 10% normal rabbit antiserum (Vector Laboratories, Burlingame, CA, USA) for 10 min. Sections were incubated with monoclonal antibodies against rat OPN (Immuno-Biological Laboratories, Gunma, Japan) using the Vectastain ABC mouse kit (Vector Laboratories) according to the manufacturer instructions. The procedure was as follows: (1) incubation with primary antibody at a dilution of 1:50; (2) incubation with 1:200 diluted secondary anti-mouse IgG antibody for 30 min; (3) incubation of the avidin-biotin-peroxidase complex (ABC; Vector Laboratories) at a dilution of 1:50 for 30 min; and (4) treatment with diaminobenzidine for color development and nuclear counterstaining with Mayer’s hematoxylin. After staining, the cells were observed using an optical microscope. The control sections were treated in the same way in the absence of the primary antibody.

### 2.7. Tartrate-Resistant Acid Phosphatase (TRAP) Staining

TRAP staining was performed on a sample collected 6 months after implantation to confirm remodeling by osteoclasts. The staining was performed using the TRAP Staining Kit (Primary Cell, Hokkaido, Japan) according to the manufacturer’s instructions.

### 2.8. Statistical Analysis

All data are presented as the mean ±SD. Statistical analysis was performed using two-sided t-tests. P values < 0.01 were considered statistically significant.

## 3. Results

### 3.1. Implantation Using Matrigel 

#### Morphological Evaluation via Micro-CT

In the control group, slight bone regeneration was observed from the excised margin. However, bone defects remained even 6 months after implantation, and continuous zygomatic arch regeneration was not observed (Figure 3, top).

In the TCP group, fusion with the zygomatic bone was observed at the anterior site 6 months after implantation, but there was no union at the posterior site. A slight gap also remained between β-TCP and the osteotomy stump. In addition, β-TCP was hardly absorbed, and the shape of the material was maintained (Figure 3, center).

In the BMP group, complete union with the zygomatic bone was observed 6 months after implantation. In addition, the β-TCP was partially absorbed and had a similar morphology to that of the zygomatic bone on the unaffected side (Figure 3, bottom).

### 3.2. Implantation Using Collagen Gel 

#### 3.2.1. Morphological Evaluation via Micro-CT

In the TCP group, a slight gap was formed 1 month after implantation, and no fusion with the zygomatic bone was observed (Figure 4a). The gap disappeared with the formation of new bone detected between the edge of the bone defect and the β-TCP within 3 months after implantation (Figure 4a). After 6 months, new bone formation was slightly increased between the bone stump and β-TCP. However, the cross-sectional image showed a slight gap remaining and no complete union was observed. The shape of β-TCP was maintained (Figure 4a,b).

In the BMP group, there was no gap between the zygomatic bone stumps and β-TCP 1 month after implantation, and complete union was observed. In addition, osteogenesis was observed connecting the zygomatic bone stumps around β-TCP (Figure 4a). After 3 months, the β-TCP was gradually absorbed at the part protruding from the zygomatic arch (Figure 4a). Six months later, β-TCP was further absorbed, and the shape resembled that of the zygomatic bone on the unaffected side. In the cross-sectional image, β-TCP and residual bone were completely united, and β-TCP was absorbed to reproduce the morphology of the zygomatic bone (Figure 4a,b).

#### 3.2.2. Detection of Volume Change and Cross-Sectional Area Change of β-TCP via Micro-CT

There was no change in volume at 1 month after implantation in both the TCP and BMP groups (*P* > 0.1).

The volume ratio of the TCP group was 99.1 ± 1.1% at 3 months and was 97.6 ± 2.8% at 6 months, showing almost no decrease in volume (*P* > 0.1). However, the volume ratio of the BMP group decreased significantly to 92.4 ± 1.2% at 3 months and to 76.4 ± 3.5% at 6 months (*P* < 0.01), which was significantly different to those of the TCP group (*P* < 0.01) (Figure 5a). 

The area change of the posterior β-TCP cross-section was 98.0 ± 0.9% in the TCP group and was 64.8 ± 1.8% in the BMP group at 6 months after implantation, showing a significant difference between the groups (*P* < 0.01) (Figure 5b).

#### 3.2.3. Histological Assessment

For assessment of early bone formation, the samples were histologically examined 1 month after implantation. In the TCP group, fibrous tissue formation was observed between the existing bone and β-TCP, but no bone tissue formation was observed (Figure 6a–c). In addition, only fibrous tissue was observed in the pores of β-TCP, but no bone tissue was observed (Figure 6d). Immunohistochemical OPN staining showed negative staining in the fibrous tissue in β-TCP pores (data not shown).

In the BMP group, bone tissue formation was observed, indicating its addition to β-TCP (Figure 7a), and a continuous image with the existing bone tissue was observed (Figure 7a–c). In addition, there were some images in which TCP was partially absorbed, and the semicircular absorbed site was replaced with bone tissue (Figure 7b,d). Bone marrow-like tissue was also formed in the area surrounded by new bone in the holes of β-TCP (Figure 7b–d). Immunohistochemical OPN staining showed positive staining on the surface of the bone tissue formed in β-TCP pores (Figure 7e,f).

For assessment of long-term bone formation and remodeling, we histologically assessed the samples collected 6 months after implantation. In the TCP group, bone tissue was partially formed in the β-TCP pores (Figure 8a–d). However, bone marrow tissue and fibrous tissue were mixed, and no formation of bone marrow-like tissue was observed (Figure 8c,d). In addition, continuity with the existing zygomatic bone was confirmed in the anterior region (Figure 8b), but fibrous tissue was found between the zygomatic bone and the new bone in the posterior region (Figure 8c). Almost all β-TCP remained, and only a few osteoclasts were observed by TRAP staining (Figure 8e,f).

In the BMP group, continuity between the existing bone and new bone in the β-TCP pore was maintained, and bone marrow-like tissue was also observed (Figure 9a–c). In β-TCP and new bone, the part protruding from the existing zygomatic bone shape was absorbed to make a worm-eaten form (Figure 9a,c), and TRAP-positive osteoclasts gathered in the same area (Figure 9e,f). We also confirmed the site where β-TCP was absorbed and replaced with bone tissue (Figure 9c,d). By contrast, β-TCP was not absorbed at the site that was continuous with the existing bone, and osteoclast migration to the site was not observed (Figure 9e).

## 4. Discussion

The treatment of zygomatic bone defect requires artificial biomaterials that induce vigorous bone formation, and most importantly, aesthetic recovery is also essential for the treatments of zygomatic bone defect. In the field of plastic surgery, it is the most important for zygomatic bone regeneration to recover bone tissue aesthetically, so simply regenerating bone tissue does not meet the needs of patient. However, to date, there have been no reports of artificial biomaterials that meet these conditions. To the best of our knowledge, the present study is the first reported attempt to form vigorous and aesthetic bone tissue in zygomatic bone defect using honeycomb β-TCP in long-term observation.

In many artificial bone studies, the materials, porosity, and pore shape have been optimized to enhance internal vascular invasion, osteocyte differentiation, and tissue formation using bioceramics such as HAp and β-TCP [13,17,18,19,20,21,22]. Some long-term studies have been conducted, but most studies have only assessed bone formation based on extraosseous implantation such as intramuscular implantation [19,20,21,22], and there are few reports on long-term bone formation and morphological changes after intraosseous implantation. In our previous study, we used honeycomb β-TCP with complete through-and-through holes with a pore size of 300 μm, which was transplanted into a rat model with a complete zygomatic bone defect. In this previous study, we confirmed good bone tissue regeneration [16]. However, this was a short-term result obtained 1 month after implantation alongside the addition of the potent bone growth factor BMP-2. Therefore, most of the β-TCP was not absorbed, and the aesthetic recovery was also not sufficient. Thus, it was necessary to observe the long-term morphological and histological changes. In addition, 1 month after implantation, bone induction was not observed with honeycomb β-TCP alone without BMP-2. However, some studies in which bioceramics were ectopically implanted showed bone formation over a long period of time, demonstrating its bone inducibility [19,20,22]. Yuan et al. [22] implanted β-TCP into the muscle of a dog and did not observe bone formation after 30 days, but it was observed after 150 days. From these reports, we postulated that bone induction may be achieved by long-term placement of honeycomb β-TCP without the addition of BMP-2.

In the previous zygomatic bone defect model study, the experiments were conducted using Matrigel as carrier for delivery of BMP-2 [16]. Although Matrigel is widely used in many bone-inducing animal experiments, it is extracted from cultured mouse osteosarcoma cells. This complication makes extrapolation of these results for human application difficult. Therefore, new experiments were performed using Matrigel and collagen gel as BMP-2 carriers in our previous skull defect model experiment using honeycomb β-TCP [23]. The results showed no significant differences in bone formation among both Matrigel group and collagen gel group. However, the skull and zygomatic bone are different regarding their histological and physiological structures. The skull has plate-like morphology and does not move physiologically, and its osteoblasts are supplied from the dura mater. In contrast, the zygomatic bone is constantly moving, since it is connected to the muscles involved in mastication, and the osteoblasts cannot be supplied by the periosteum in a completely transected bone defect. Thus, it is more difficult to regenerate zygomatic bone. To test this possibility, in the present study, we first conducted experiments using Matrigel as a carrier for growth factors. In the control group, only partial bone regeneration was obtained at 6 months, and the defect remained. In the TCP group, bone formation that could not be confirmed after 1 month could be confirmed after 6 months. This result is consistent with that obtained by Yuan et al. [22]. In the BMP group, good union with the bone was achieved, and remodeling was observed wherein the shape of the implanted β-TCP changed to a morphology very similar to that of the original zygomatic bone. Next, we conducted experiments using collagen gel as a carrier for BMP-2. One month after the implantation, β-TCP with BMP-2 using the collagen gel also showed good bone union, similar to the result of a previous study, in which Matrigel was used as a carrier [16]. Furthermore, to observe the process of long-term bone formation and morphological changes, micro-CT follow-up was performed over time at 1, 3, and 6 months after the implantation. In the BMP group, bone formation similar to the morphology of the zygomatic bone was observed around β-TCP at 1 month. Six months later, it was confirmed that the β-TCP and new bone were absorbed, and remodeling occurred to reproduce the original zygomatic bone shape. In the TCP group, bone formation was observed in both the stump of the β-TCP and in the central part of the β-TCP pore. This phenomenon may indicate that bone induction was obtained without BMP-2. In this study, there was no major difference of bone formation and remodeling between Matrigel and collagen gel in CT images and histological findings at 1 and 6 months (data not shown), demonstrating the usefulness of the honeycomb β-TCP using a clinically applicable collagen gel.

Interestingly, in the BMP group, new bone was formed on the entirety of the β-TCP at first. Then, bone resorption occurred in the part protruding from the original zygomatic bone shape, and the zygomatic bone shape was naturally reproduced. In other words, bone resorption occurred due to remodeling. In clinical practice, some reports have indicated that hypertrophy or atrophy of the vascularized fibular bone graft is caused by remodeling after reconstruction of the lower extremity or the mandible [24,25,26,27,28]. The study in which vascularized bone grafts were implanted in rats and examined with or without mechanical stress showed that mechanical stress induced bone hypertrophy; conversely, reduced mechanical stress resulted in a decreased bone mass [29,30]. Similar to these reports, the honeycomb β-TCP used in this study was also incorporated into the bone matrix or gradually replaced with bone tissue via remodeling at the site of mechanical stress. At the site protruding from the original zygomatic bone shape, we postulated that β-TCP was absorbed via remodeling to restore the original zygomatic bone shape because there was no mechanical stress.

In the TCP group, the stump of the zygomatic bone was absorbed after 1 month, after which bone formation occurred from the stump and united with β-TCP. This resorption may have been caused by moving the zygomatic bone with the movement of the rat itself, because a rat is not able to rest spontaneously. Therefore, if the zygomatic bone can be internally fixed and kept at rest, good bone regeneration may be achieved from an early stage.

The main limitation of this study was that the surgical site could not be kept at rest after the operation. Moreover, the defect was a small area because β-TCP was implanted in the zygomatic bone of the rat. If β-TCP is implanted in large animals and internal fixation leads to the surgical site being at rest, good bone formation may be obtained without the addition of BMP-2. In addition, this effect can be confirmed in a state closer to the clinical state by implanting in a larger defect. Although the observation period in this study was 6 months, further morphological changes may be observed by carrying out analyses for a longer period. In addition, more frequent follow-up may lead to the elucidation of new mechanisms of remodeling. Finally, in the future, in order to elucidate the mechanism of TCP remodeling in bone reconstruction, we would like to examine the factors related to mechanical stress, such as MAPK signaling pathway and so on, which has been reported to be related to bone remodeling in recent years [31,32].

## 5. Conclusions

We demonstrate that honeycomb β-TCP alone may not only improve bone conduction but also bone induction. We also showed that honeycomb β-TCP with added BMP-2 has excellent bone inducibility using clinically applicable collagen gel, and that remodeling of new bone may be obtained in the long term.

This honeycomb β-TCP may be very useful in a clinical setting as a new bone reconstruction material, providing an alternative to vascularized bone grafts.

## Figures and Tables

**Figure 1 materials-13-05374-f001:**
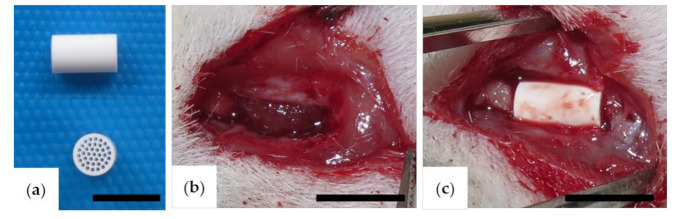
(**a**) Honeycomb β-tricalcium phosphate (TCP). Intraoperative photos of the (**b**) defect (5 mm) made at the rat zygomatic bone and the (**c**) implanted honeycomb β-TCP material. Scale bars = 5 mm.

**Figure 2 materials-13-05374-f002:**
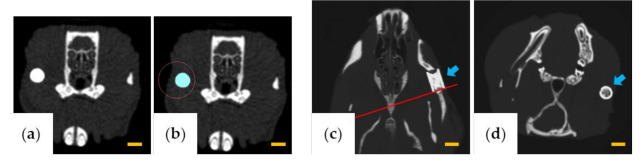
Images for measuring the β-tricalcium phosphate (TCP) volume and β-TCP cross-sectional area. (**a**,**b**) Images used for volume measurement. (**a**) Computed tomography (CT) image of the area containing β-TCP and (**b**) designation of the region of interest. (**c,d**) Images used for cross-sectional area measurement. (**c**) β-TCP long-axis plane (red line indicates the rear end position). (**d**) Plane perpendicular to the through-holes at the position of the red line in (**c**). The arrows indicate β-TCP. Scale bars = 3 mm.

**Figure 3 materials-13-05374-f003:**
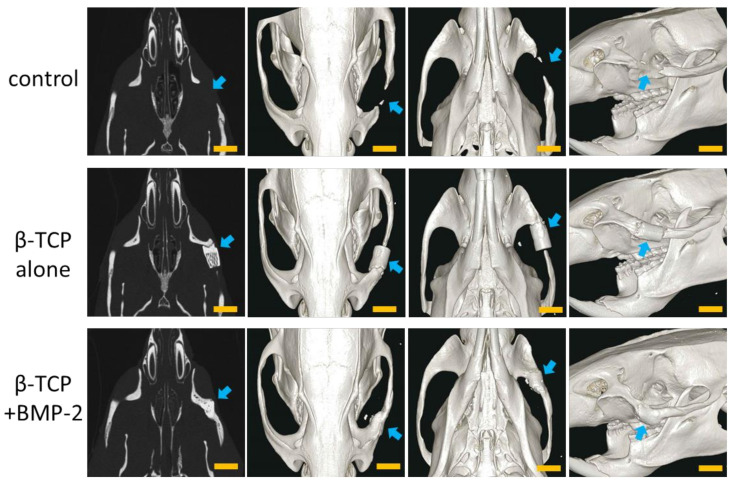
Micro-computed tomography (CT) images of the heads of rats at 6 months after the operation using Matrigel. **Top**: control group; **center**: β -tricalcium phosphate (TCP) alone group (TCP group); **bottom**: β -TCP + BMP-2 group (BMP group). The arrows indicate the surgical sites. Scale bars = 5 mm.

**Figure 4 materials-13-05374-f004:**
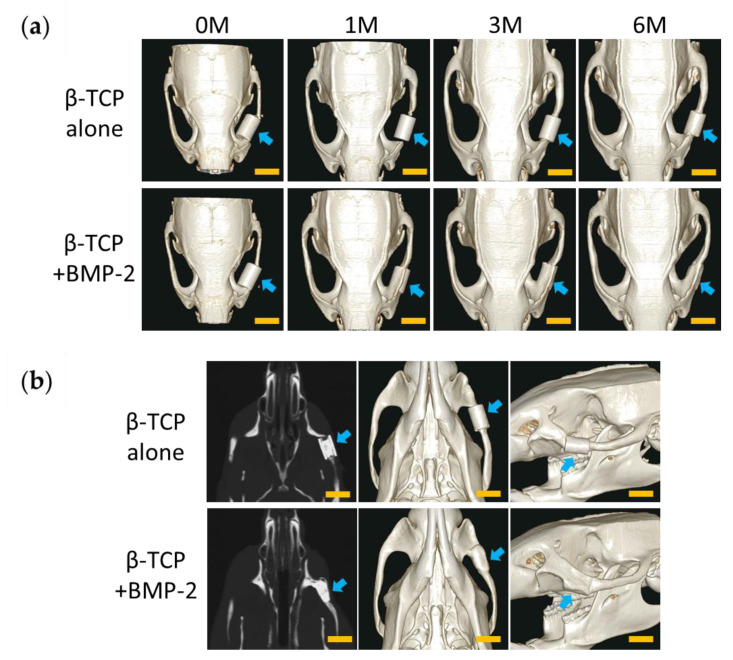
Micro-computed tomography (CT) after implantation with collagen gel. (**a**) Three-dimensional (3D) reconstruction image, from left to right: immediately after the operation, 1 month after the operation, 3 months after the operation, and 6 months after the operation [top: β-tricalcium phosphate (TCP) alone group (TCP group), bottom: β-TCP + BMP-2 group (BMP group)]. (**b**) Detailed images after 6 months (left: cross-section image; center and right: 3D images from other angles; top: TCP group; bottom: BMP group). Arrows indicate surgical sites. M: month. Scale bars = 5 mm.

**Figure 5 materials-13-05374-f005:**
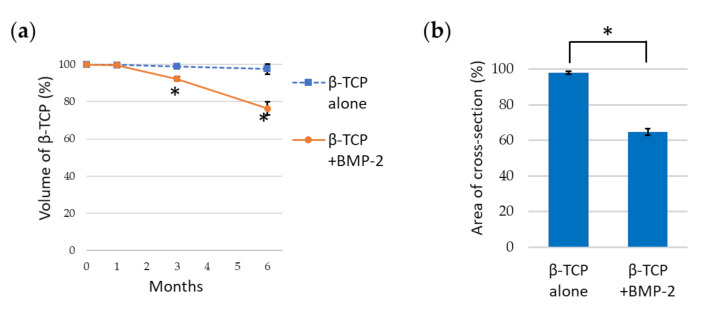
(**a**) Changes in the volume of β-tricalcium phosphate (TCP). (**b**) Percentage of the posterior cross-sectional area 6 months after implantation. Data are shown as the mean ±SD. * *P* < 0.01.

**Figure 6 materials-13-05374-f006:**
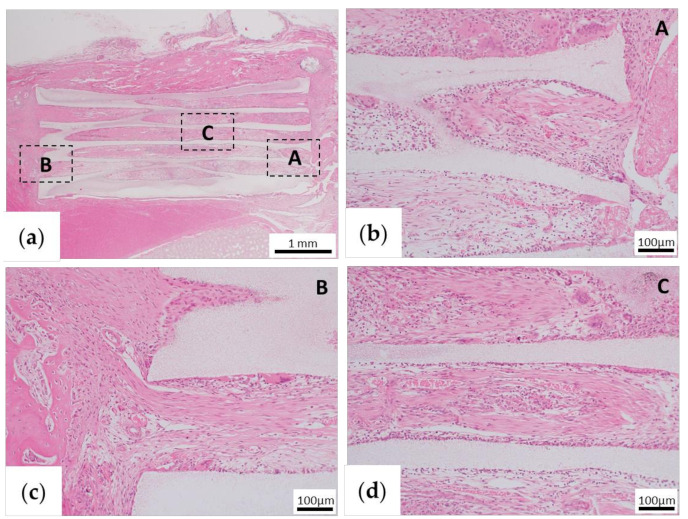
Histological findings of the β -tricalcium phosphate (TCP) alone group (TCP group) 1 month after implantation. (**a**) Low-magnification image of hematoxylin and eosin (HE) staining. (**b–d**) High-magnification images of the parts that are framed in (**a**). (**b,c**) The fibrous tissue is continuous from the end of β-TCP to the inside of the pore. (**c**) Fibrous tissue formed between the stump of the existing bone and β-TCP. (**d**) In the central part of the β-TCP pore, only fibrous tissue formation is observed, and there is no obvious bone tissue formation.

**Figure 7 materials-13-05374-f007:**
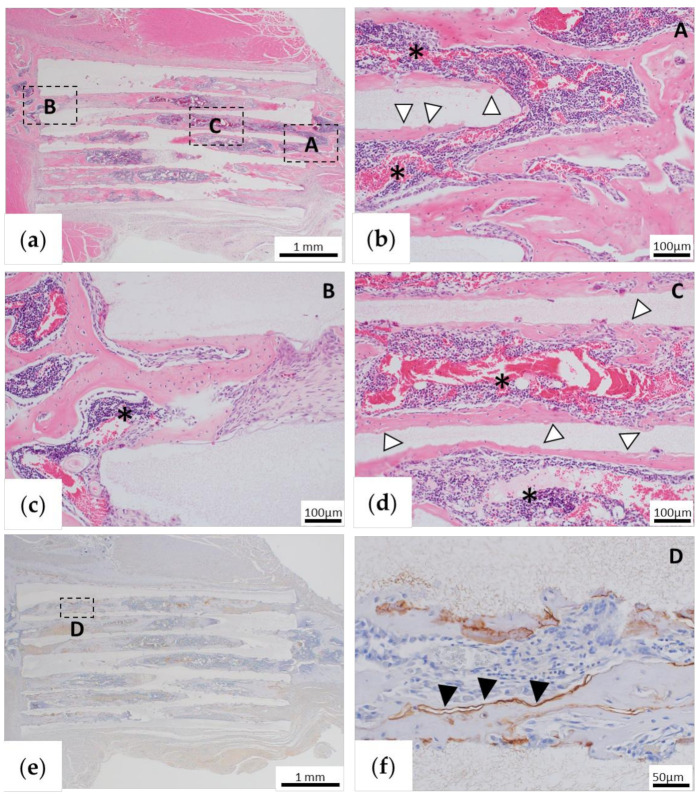
Histological findings of the β-tricalcium phosphate (TCP) + BMP-2 group (BMP group) 1 month after implantation. (**a**) Low-magnification image of hematoxylin and eosin (HE) staining. (**b–d**) High-magnification images of the parts that are framed in (**a**). (**b,c**) Continuous invasion of bone cells and bone marrow-like tissue can be observed from the stump of the zygomatic bone into the hole of β-TCP. (**d**) Bone cells are observed joining the inner wall of the pore to the central part of β-TCP, and bone marrow-like tissue formation is observed inside. (**e**) Low-magnification image of osteopontin immunostaining. (**f**) High-magnification images of the parts that are framed in (**e**). Asterisks are bone marrow-like tissue formed in β-TCP pores. White arrowheads indicate the site where β-TCP is replaced by bone cells. Black arrowheads are osteopontin-positive sites.

**Figure 8 materials-13-05374-f008:**
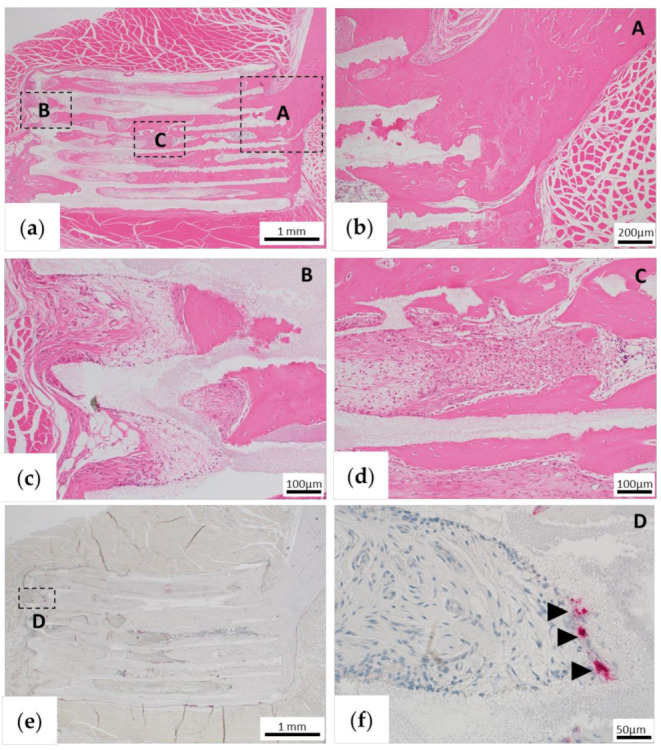
Histological findings in the β-tricalcium phosphate (TCP) alone group (TCP group) 6 months after implantation. (**a**) Low-magnification image of hematoxylin and eosin (HE) staining. (**b**–**d**) High-magnification images of the parts that are framed in (**a**). (**b**) Bone tissue formation is continuous with existing bone. (**c**) Bone formation is observed in the foramen, but fibrous tissue is present at the junction between β-TCP and existing bone, and there is no continuity of bone tissue with the existing bone stump. (**d**) Bone tissue formation is observed along the TCP wall, but fibrous tissue is observed inside. (**e**) Low-magnification image of tartrate-resistant acid phosphatase (TRAP) staining. (**f**) High-magnification images of the parts that are framed in (**e**). Black arrowheads indicate TRAP-positive osteoclasts.

**Figure 9 materials-13-05374-f009:**
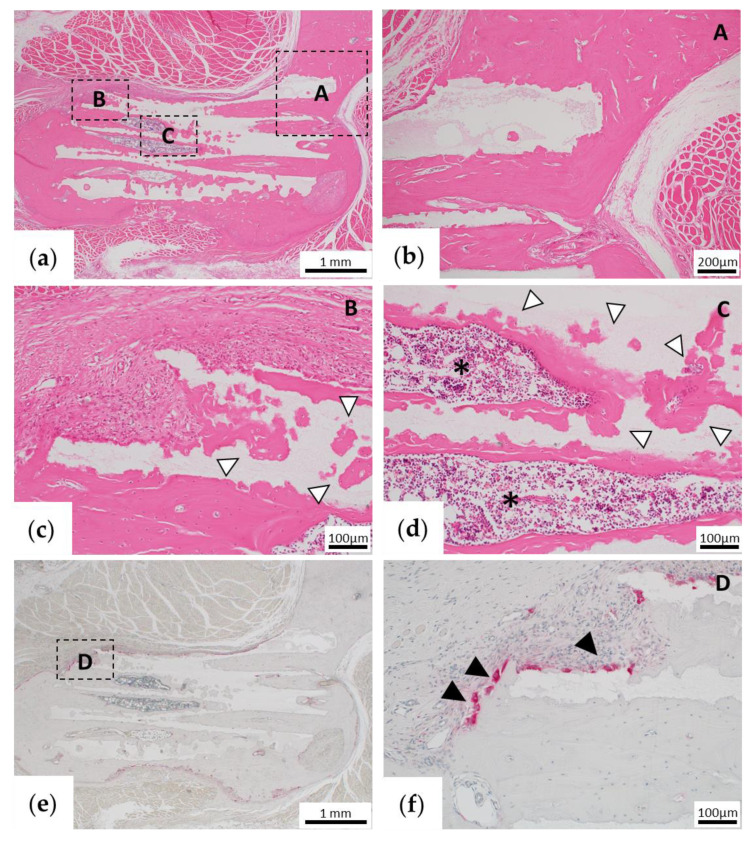
Histological findings of the β-tricalcium phosphate (TCP) + BMP-2 group (BMP group) 6 months after implantation. (**a**) Low-magnification image of hematoxylin and eosin (HE) staining. (**b**–**d**) High-magnification images of the sections indicated in (**a**). (**b**) Bone formation is observed in the β-TCP hole, which is continuous with the existing bone. (**c**) Absorption of β-TCP is observable. (**d**) Bone tissue formation is observed along the wall in the β-TCP hole. Bone marrow-like tissue can be observed inside. (**e**) Low-magnification image of tartrate-resistant acid phosphatase (TRAP) staining. (**f**) High-magnification images of the sections indicated in (**e**). Osteoclast collection is observed in the β-TCP absorption area. Asterisks indicate bone marrow-like tissue formed in β-TCP pores. White arrowheads indicate the site where β-TCP is replaced by bone cells. Black arrowheads indicate TRAP-positive osteoclasts.

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
