# Peer review of "Long-Term Effect of Honeycomb β-Tricalcium Phosphate on Zygomatic Bone Regeneration in Rats"

_materials, 2020, doi:10.3390/ma13235374_

Round 1
Reviewer 1 Report
A well-conducted study describing the promising bioceramic material, which demonstrated excellent osteoconductivity, osteoinductive featurs, and biocompatibility. The paper deserves publication after the correction of the following points:
Minor:
- M&M, line 176: please change "Meyer's" to "Mayer's" hematoxylin.
- In the Discussion, please describe the advantages and shortcomings of the selected zygomatic bone replacement model and other canonical models used to assess bone regeneration (e.g., calvarial defect model). This can be incorporated into the paragraph describing the differences across the two models (lines 349-359).
- Lines 355-356: "however, it is noteworthy that in this study, β-TCP was implanted in a site where rest can be maintained"... Did you mean the study [23] or that performed in the article?
- Why did you separate osteopontin and TRAP stainings to different experiments? Please explain in the Discussion. Please also describe another possible stainings to properly examine the bone remodeling in the limitations of the study (lines 373-380).
Author Response
Thank you very much for reviewing our manuscript and offering valuable advice. We are thankful for the time and energy you expended. Our responses to your comments are as follow:
- M&M, line 176: please change "Meyer's" to "Mayer's" hematoxylin.
→We have corrected the text. (Line 179)
- In the Discussion, please describe the advantages and shortcomings of the selected zygomatic bone replacement model and other canonical models used to assess bone regeneration (e.g., calvarial defect model). This can be incorporated into the paragraph describing the differences across the two models (lines 349-359).
→We have modified and added to the text. (Line 364-369)
- Lines 355-356: "however, it is noteworthy that in this study, β-TCP was implanted in a site where rest can be maintained"... Did you mean the study [23] or that performed in the article?
→”This study” presented the previous skull defect study. We have changed “this” to “our previous”.
- Why did you separate osteopontin and TRAP stainings to different experiments? Please explain in the Discussion. Please also describe another possible stainings to properly examine the bone remodeling in the limitations of the study (lines 373-380).
→At 1 month, we wanted to investigate the bone formation in TCP holes, so we have stained osteopontin. On the other hands, at 6 months, we wanted to investigate the TCP resorption and remodeling TCP rather than bone formation, thus we have stained TRAP. We have mentioned this points in Materials and Methods (Line 168-169, 184-185). For bone remodeling, RANKL or CD68 staining can be considered, but TRAP is the most common and optimal. However, further studies are needed to elucidate the mechanism of bone remodeling of TCP. Thus, we have mentioned the limitations of this study (Line 390-393).
Reviewer 2 Report
This work is a continuation of their previous study on the short-term evaluation of TCP scaffolds in vivo. The study was well designed and implemented, and the results are of significance to the bioceramics and orthopaedic implants research community. The following aspects should be addressed prior to the manuscript’s acceptance for publication:
- What were the differences in osteoconductivity and osteoinductivity, if any, between the collagen/TCP scaffold and the Matrigel/TCP scaffolds? What were the reasons for the differences if any?
- Sections 2.3, 2.31, 2.4, 2.41 – these sections can be condensed to reduce redundant, overlapping descriptions.
- Please explain how the amount of BMP2 was determined/optimized for loading in the scaffolds.
- The use of BMP2 for bone healing is not without drawbacks. Side effects include soft tissue swelling, ectopic bone formation, and resorption of adjacent bone. It was also reported that BMP2 at a concentration of 50 ng/mL can induce apoptosis of human osteoblasts. Please include comments on such possible side effects and if any of these side effects were observed in this study.
- Figure 6 – has OPN or any other bone markers been stained?
- Compared to OPN, a more commonly used and well recognized bone marker is OCN. Please justify the use of OPN but not OCN.
- The authors claimed that the TCP scaffolds may offer an alternative bone reconstruction material to vascularized bone grafts. The ability to induce vascularization has been recognized as an important aspect of bone implants. Has any work been done to assess the vascularization induced by the TCP scaffolds?
- Figure 1 – 4, please add scale bars to the figures.
Author Response
Thank you very much for reviewing our manuscript and offering valuable advice. We are thankful for the time and energy you expended. Our responses to your comments are as follow:
- What were the differences in osteoconductivity and osteoinductivity, if any, between the collagen/TCP scaffold and the Matrigel/TCP scaffolds? What were the reasons for the differences if any?
→There was no significant difference in osteoconductivity and osteoinductivity in between the collagen/TCP scaffolds and Matrigel/TCP scaffolds because there was no clear difference in CT images and histological findings in between the collagen/TCP scaffolds and Matrigel/TCP scaffolds at 1 and 6 months (data not shown). We have added to the text (Line 353-354).
- Sections 2.3, 2.31, 2.4, 2.41 – these sections can be condensed to reduce redundant, overlapping descriptions.
→We have modified the materials and methods.
- Please explain how the amount of BMP2 was determined/optimized for loading in the scaffolds.
→We have explained the determination of BMP-2 concentration (Line 90-91).
- The use of BMP2 for bone healing is not without drawbacks. Side effects include soft tissue swelling, ectopic bone formation, and resorption of adjacent bone. It was also reported that BMP2 at a concentration of 50 ng/mL can induce apoptosis of human osteoblasts. Please include comments on such possible side effects and if any of these side effects were observed in this study.
→Side effects from BMP-2 did not occur in this experiments and our previous studies. The BMP-2 concentration required to induce bone formation is rodent (0.02-0.4 mg/ml), non-human primate (0.75-2.0 mg/ml), and FDA approved concentration is 1.5 mg/ml. Thus, the concentration of BMP-2 in our study was within the appropriate rang in the literatures (J Bone Joint Surg Am. 2010;92:411-426, Tissue Eng Part B. 2016;22:284-297).
- Figure 6 – has OPN or any other bone markers been stained?
→We have stained OPN. We have added to the text (Line 256-257).
- Compared to OPN, a more commonly used and well recognized bone marker is OCN. Please justify the use of OPN but not OCN.
→We agree the more commonly bone marker is OCN. However, in this experiment, OPN was more appropriate than OCN because Figure 7 experiment was in the middle stage of bone formation, which was one month, osteopontin expressed in the middle stage of bone formation was more appropriate than osteocalcin which expressed in the late stage.
- The authors claimed that the TCP scaffolds may offer an alternative bone reconstruction material to vascularized bone grafts. The ability to induce vascularization has been recognized as an important aspect of bone implants. Has any work been done to assess the vascularization induced by the TCP scaffolds?
→In our previous study, we have investigated vascularization of honeycomb TCP in ectopic bone formation (Int J Med Sci. 2018, 15(14),1582-1590). However, we have not examined vascularization of honeycomb TCP in zygomatic bone defect model. Therefore, we will consider to stain VEGF or CD34 and so on in the future.
- Figure 1 – 4, please add scale bars to the figures.
→We have added to the scale bars in Figure1-4.
Reviewer 3 Report
The manuscript by Nakagiri R. et al. reports a study on the honeycomb β-tricalcium phosphate on zygomatic bone regeneration in rats. The results are a continuation of research published in Journal of Biomedical Materials Research 2014. In presented work Authors focused on one type of honeycomb β-TCP (diameter of 3 mm, length of 5 mm, 37 through-and-through holes with 300-μm pores) impregnated with bone morphogenic protein (BMP)-2, diluted with Matrigel or collagen gel. Long-term observation has been performed, up to 6 months. Despite the results demonstrate that honeycomb β-TCP with added BMP-2 has good bone inducibility and has promoted remodelling of new bone, the presented materials (β-TCP + BMP-2 in Matrigel or collagen gel, conc. 80 µg/ml) are already published - Materials 2020, 13(21), 476. Please explain in details why the change of substituted bone (skull vs. zygomatic bone) or scaffolds dimensions (holes diameter is the same) is so critical since both papers are related to the same composition of the scaffold otherwise it looks like lack of the novelty.
Author Response
Thank you very much for providing important comments. We are thankful for the time and energy you expended. Our responses to your comments are as follow:
The manuscript by Nakagiri R. et al. reports a study on the honeycomb β-tricalcium phosphate on zygomatic bone regeneration in rats. The results are a continuation of research published in Journal of Biomedical Materials Research 2014. In presented work Authors focused on one type of honeycomb β-TCP (diameter of 3 mm, length of 5 mm, 37 through-and-through holes with 300-μm pores) impregnated with bone morphogenic protein (BMP)-2, diluted with Matrigel or collagen gel. Long-term observation has been performed, up to 6 months. Despite the results demonstrate that honeycomb β-TCP with added BMP-2 has good bone inducibility and has promoted remodelling of new bone, the presented materials (β-TCP + BMP-2 in Matrigel or collagen gel, conc. 80 µg/ml) are already published - Materials 2020, 13(21), 476. Please explain in details why the change of substituted bone (skull vs. zygomatic bone) or scaffolds dimensions (holes diameter is the same) is so critical since both papers are related to the same composition of the scaffold otherwise it looks like lack of the novelty.
→The honeycomb β-TCP with 300-μm through-and-through holes induced the most vigorous bone formation in our previous study (reference 16). We have mentioned this point in Introduction. Next, in skull defect model (reference 23) and zygomatic bone defect model (in this study), we used the same honeycomb β-TCP with 300-μm through-and-through holes as a bone reconstruction material.
We have mentioned the novelty of this study in Discussion (Line364-369).
Round 2
Reviewer 3 Report
The Authors of reviewed paper Long-term effect of honeycomb β-tricalcium 3 phosphate on zygomatic bone regeneration in rats Ryoko Nakagiri et al. indicate that the novelty of presented paper is described in following part:
Page 12, lines: 364-369: “Although Matrigel is widely used in many bone-inducing animal experiments, it is extracted from cultured mouse osteosarcoma cells. This complication makes extrapolation of these results for human application difficult. Therefore, we used a collagen gel, which can be applied to humans as an alternative carrier in this study. One month after the implantation, β-TCP with BMP-2 using the collagen gel also showed good bone union, similar to the result of a previous study, in which Matrigel was used as a carrier [16].”
The same argumentation is presented in following paper of the group, which is already accepted to the same Special Issue "Materials for Hard Tissue Repair and Regeneration" of Materials journal:
Effect of Honeycomb β-TCP Geometrical Structure on Bone Tissue Regeneration in Skull Defect Materials 2020, 13, 4761 by Toshiyuki Watanabe , Kiyofumi Takabatake, Hidetsugu Tsujigiwa , Satoko Watanabe, Ryoko Nakagiri , Keisuke Nakano , Hitoshi Nagatsuka and Yoshihiro Kimata
Page 10, lines 11-16: “Matrigel is widely used as a carrier for growth factors in bone tissue regeneration experiments, and Matrigel has BMP-2 retention and has been used in many osteoinductive animal experiments. Like other experiments, vigorous bone formation was observed in the β-TCP holes in this experimental setup using Matrigel. However, Matrigel is extracted from animals (mouse osteosarcoma), and thus clinical application in humans is actually difficult. Therefore, we also used Collagen gel, which is more applicable to humans.”
Both papers present the same material and analogous modification, the only difference is the geometry of the scaffolds and bone type which has been replaced. Also, the methodology of discussed results are comparable. Therefore, in my opinion the presented results exhibit restricted novelty to be published in Materials journal.
Round 3
Reviewer 3 Report
Thank you for extending of your manuscript, I accept it in present form.